# Estimation of Genetic Parameters for Early Growth Traits in Luzhong Mutton Sheep

**DOI:** 10.3390/ani14121754

**Published:** 2024-06-10

**Authors:** Yifan Ren, Xue Li, Junmin He, Menghua Zhang, Guifen Liu, Chen Wei, Guoping Zhang, Wenhao Zhang, Fumei Nie, Ming Wang, Kechuan Tian, Xixia Huang

**Affiliations:** 1Institute of Animal Science and Veterinary Medicine, Shandong Academy of Agricultural Sciences, Jinan 250100, China; 15136225372@163.com (Y.R.); lixueli1126@163.com (X.L.); hejunmin330@163.com (J.H.); liuguifen126@126.com (G.L.); weichenchen1989@126.com (C.W.); gpzhangsaas@163.com (G.Z.); zwh22635@163.com (W.Z.); niefumei@163.com (F.N.); 2College of Animal Science, Xinjiang Agricultural University, Urumqi 830052, China; zhangmenghua810@126.com (M.Z.); xjming2024@163.com (M.W.)

**Keywords:** animal model, early growth traits, genetic parameters, maternal effects, Luzhong mutton sheep

## Abstract

**Simple Summary:**

Luzhong mutton sheep, a new breed of sheep used for meat production, are characterized by a fast growth rate, high adaptability, and roughage tolerance. The aim of this study was to estimate the genetic parameters and analyze the genetic trends of early growth traits in Luzhong mutton sheep using different animal models, and Model 4 was identified as the best model. In this study, the greater heritability of weaning weight and higher genetic correlation with key production traits indicated that weaning weight could be used as a selection criterion for late weight traits. In addition, the results of this study indicated that maternal effects had a great influence on early growth traits in Luzhong mutton sheep.

**Abstract:**

In this study, six different animal models were fitted, and the constrained maximum likelihood method was used to assess the genetic parameters and genetic trends of early growth traits in Luzhong mutton sheep. The experimental data of this study included the newborn weight (BWT, N = 2464), weaning weight (WWT, N = 2923), weight at 6 months of age (6WT, N = 2428), average daily weight gain from birth to weaning (ADG1, N = 2424), and average daily weight gain from weaning to 6 months of age (ADG2, N = 1836) in Luzhong mutton sheep (2015~2019). The best model for the genetic parameters of the five traits in Luzhong mutton sheep was identified as Model 4 using the Akaike information criterion (AIC) and likelihood ratio test (LRT) methods, in which the estimated values of direct heritability for the BWT, WWT, 6WT, ADG1, and ADG2 were 0.156 ± 0.057, 0.547 ± 0.031, 0.653 ± 0.031, 0.531 ± 0.035, and 0.052 ± 0.046, respectively, and the values for maternal heritability were 0.201 ± 0.100, 0.280 ± 0.047, 0.197 ± 0.053, 0.275 ± 0.052, and 0.081 ± 0.092, respectively. The genetic correlation between the ADG2 and WWT was negative, and the genetic and phenotypic correlations among the remaining traits were positive. In this study, maternal effects had a more significant influence on early growth traits in Luzhong mutton sheep. In conclusion, to effectively improve the accuracy of genetic parameter estimation, maternal effects must be fully considered to ensure more accurate and better breeding planning.

## 1. Introduction

China ranks first in the world in terms of sheep farming, with 194.03 million sheep in 2022 [1]. Livestock growth is influenced by a combination of genetic and environmental factors, and high-performance growth traits usually lead to higher productivity and better economics, which in turn positively affect the income of farmers [2,3]. Therefore, farmers usually select livestock with good growth traits for breeding. The growth potential of lambs determines their later meat production capacity, which is one of the most important factors affecting economic efficiency. Early selection of lambs based on scientific breeding programs can effectively increase production profit and improve breeding efficiency. The accurate estimation of heritability and the correlation of growth traits is very important for the development of effective genetic improvement programs [4]. Genetic parameter estimation requires a rational mathematical model that not only includes the various genetic and environmental factors affecting the trait but is also easy to manipulate and realize in production practice [5]. The growth performance of livestock is affected not only by direct additive genetic effects but also by maternal genetic effects and maternal permanent environmental effects. The results of previous studies have shown that [6,7] the incorporation of maternal effects into the analytical model allowed more accurate estimations of the genetic parameters of the trait.

Luzhong mutton sheep is a new breed of meat sheep that produces high-quality and high-grade mutton using South African White-headed Dorper sheep as the sire and Hu sheep from China as the dam; this sheep was bred in 2020 and included in the list of national livestock and poultry genetic resources in 2021 [8]. The breed was bred by 11 units, including Jinan Laiwu Yingtai Agricultural and Animal Husbandry Science and Technology Co., Ltd. and the Beijing Animal Husbandry and Veterinary Research Institute of the Chinese Academy of Agricultural Sciences, over a period of 14 years. The Luzhong meat sheep ewes are in heat all year round, with three births in two years and an average lambing rate of 231.83%. Luzhong mutton sheep are mainly distributed in Shandong Province and have a white coat, fast growth rate, roughage tolerance, strong disease resistance, and good meat production performance. More importantly, the breed has a high reproductive rate and stable genetic performance and is suitable for confinement. The number of groups in the core production area was 5000 at the time of breeding, and the number of groups reached 46,000 at the time of applying for the validation of new national livestock and poultry breeds of Luzhong mutton sheep in 2019. The stock of Luzhong mutton sheep in the core production area was about 12,000 as of July 2023, and it was successively promoted to 20 provinces, municipalities, and autonomous regions of China. Luzhong mutton sheep has all demonstrated good adaptability. To further improve the production performance of Luzhong mutton sheep, it is important to select the best animal model to accurately evaluate the genetic parameters of early growth traits. This is necessary to optimize the current breeding program to select and breed on the basis of early growth traits more accurately, thereby comprehensively improving the genetic quality of the Luzhong mutton sheep population. Due to the newness of this breed, no reports on the optimal animal model for the estimation of the genetic parameters of early growth traits in Luzhong mutton sheep exist.

The objective of this study was to estimate the genetic parameters of early growth traits of Luzhong mutton sheep and explore the effects of maternal effects on each trait. In addition, we compared and analyzed the differences in the estimation effects of different models to determine the optimal model for early growth traits and improve the accuracy of genetic parameter estimation, with the aim of providing a scientific theoretical basis for adjusting and optimizing the breeding program of the Luzhong mutton sheep flock, formulating a selection program for early selection and breeding.

## 2. Materials and Methods

The experimental data for this study included five early growth traits of Luzhong mutton sheep raised at the Yingtai Organic Agriculture Technology Development Co., Ltd. in Jinan City, Shandong Province from 2015 to 2019, including the birth weight (BWT), weaning weight (WWT), 6-month-old weight (6WT), average daily weight gain from birth to 3 months (ADG1), average daily weight gain from 3 months to 6 months (ADG2), and pedigree data. The farm is located at 117°60′, 36°30′ E and has an average elevation of 117 m with a temperate monsoon climate. Winters are cold and dry, springs are warm and windy, summers are hot and rainy, and autumns are cool and sunny. The annual maximum temperature is 36.7 °C, the annual minimum temperature is −14.5 °C, and the average annual precipitation is 707.9 mm. Sheep flocks are intensively managed and house fed, according to sex and age. Sheep are divided into breeding rams and production ewes as well as reserve rams and reserve ewes. Keepers usually feed animals at different stages of life according to a certain dietary formula (Appendix A). The method of insemination of ewes is artificial insemination. Lambing is prepared for in advance, and newborn lambs are weighed within 0.5 h of birth, as soon as the wool dries and before their first feeding and kept with their mothers until weaning at 3 months of age. The descriptive statistics of the collected data, excluding missing values of sire and dam numbers, inaccurate phenotypic values, and data outside the mean ± 3 times the standard deviation, are presented in Table 1.

The growth trait data were divided into fixed effects: sex (2 levels: male and female), year of birth (5 levels: 2015, 2016, 2017, 2018, and 2019) and season of birth (4 levels: spring, summer, autumn, and winter). Birth rank and rearing rank are imperfectly classified on farms and therefore not taken into consideration. Least squares analyses were performed using the GLM procedure with SAS 8.1 software, and Duncan’s multiple comparisons method was used to further analyze the within-group differences in early growth traits of Luzhong mutton sheep and to obtain the least squares means and multiple comparisons. The linear model used is shown below:$$y_{ijkl}=\mu +x_{i}+n_{j}+s_{k}+e_{ijkl}$$
where $y_{ijkl}$ is the individual observation, μ is the population mean, $x_{i}$ is the sex effect at the ith level, $n_{j}$ is the birth year effect, $s_{k}$ is the birth season effect, and $e_{ijkl}$ is the residual effect. Animal models are superior from both a statistical and genetic breeding point of view, making fuller use of individual phenotypic and genealogical information. Therefore, to determine the optimal model for early growth traits of Luzhong mutton sheep, the following six single-trait animal models were fitted, in which different combinations of direct additive genetic effects, maternal genetic effects, and maternal permanent environmental effects were considered. Models 1, 2, 3, 4, 5, and 6 all contain fixed effects, random effects, and residual effects, and the difference lies in the different fixed effects considered in each model: Model 1 only considers direct additive genetic effects; Model 2 considers direct additive genetic effects and maternal permanent environment effects; Models 3 and 4 both consider direct additive genetic effects and maternal genetic effects, but there is a genetic covariance between the two in Model 4, not in Model 3; and Models 5 and 6 both consider direct additive genetic effects, maternal genetic effects, and maternal permanent environment effects, but there is a genetic covariance between direct additive genetic effects and maternal genetic effects in Model 6 and not in Model 5. The DMUAI module of DMU software (v6.0) was used to estimate the genetic parameters for the early growth traits of Luzhong mutton sheep [9] as reported below:(1)$$y=Xb+Z_{1}a+e\;$$
(2)$$y=Xb+Z_{1}a+Z_{3}c+e$$
(3)$$y=Xb+Z_{1}a+Z_{2}m+e\;\;C\mathrm{ov}(a,m)=0$$
(4)$$y=Xb+Z_{1}a+Z_{2}m+e\;\;C\mathrm{ov}(a,m)\neq 0$$
(5)$$y=Xb+Z_{1}a+Z_{2}m+Z_{3}c+e\;\;C\mathrm{ov}(a,m)=0$$
(6)$$y=Xb+Z_{1}a+Z_{2}m+Z_{3}c+e\;\;C\mathrm{ov}(a,m)\neq 0$$
where *y* is the vector of observations for each trait; *b* is the vector of fixed effects; *a* is the vector of direct additive genetic effects; *m* is the vector of maternal genetic effects; *c* is the vector of maternal permanent environmental effects; *e* is the vector of residual effects; and *X*, *Z*_1_, *Z*_2_, and *Z*_3_ are the correlation matrices for the fixed, direct additive genetic effects, maternal genetic effects, and maternal permanent environmental effects, respectively. The total heritability not only reflects the importance of maternal genetic effects but also measures the role of the phenotypic value of the trait in selection. The total heritability ($h_{T}^{2}$) was calculated for each model using formula [10]:
$$h_{T}^{2}=\frac{\left(\sigma_{a}^{2}+0.5\sigma_{m}^{2}+1.5\sigma_{a,m}\right)}{\sigma_{p}^{2}}$$
where $h_{T}^{2}$ is the total heritability, $\sigma_{\alpha }^{2}$ is the direct additive genetic variance, $\sigma_{m}^{2}$ is the maternal genetic variance, *σ_a,m_* is the covariance between the direct additive genetic effect and maternal genetic effect, and *σ_p_*^2^ is the phenotypic variance. The estimated effects of different animal models were examined using the Akaike information criterion (AIC) method and likelihood ratio test (LRT) method to determine the best model for estimating the genetic parameters of early growth traits in Luzhong mutton sheep. The formula for the AIC is as follows [11]:AIC=2k−2log⁡L
where *L* is the value of the maximum likelihood function and *k* is the number of parameters to be estimated. The AIC reflects the effect of the number of parameters to be estimated in the model on the estimation effect. The smaller the AIC is, the better the variance component estimation achieved [12]. Therefore, the model with the smallest AIC among a set of models is selected as the optimal model. The likelihood ratio test (LRT) was used to compare different models using the following the test statistic:$$\begin{array}{cc} LR & =-2\ln\frac{L_{MAX}|model1}{L_{MAX}|model2}\\ & =[-2\ln(L_{MAX}model1)]-[-2\ln(L_{MAX}model2)] \end{array}$$
where *LR* is the likelihood ratio and *L_MAX_model*1 and *L_MAX_model*2 are the maximum likelihood ratios under 2 different models. Model 1 is a simplified model of Model 2. *LR* approximates the chi-square distribution, and the degrees of freedom are the difference between the number of parameters estimated using Models 2 and 1. According to the critical value table of the chi-square distribution, if the difference is significant, the increased parameters have a significant effect on the model, and vice versa, which means that the effect on the model is not significant [12]. Estimated breeding value (EBV) means for each trait were calculated for lambs born from 2015 to 2019 using DMU software to obtain a genetic trend plot of EBV means for each trait as a function of birth year.

## 3. Results

### 3.1. Fixed Effects Analysis

According to the least squares analysis, three non-random factors, including the year of birth, season of birth, and non-random factors, had highly significant effects (*p* < 0.01) on all weight traits and therefore needed to be used as fixed effects in the genetic parameter estimation model for subsequent genetic analyses (Table 2).

### 3.2. Variance Components and Genetic Parameter Estimation

The variance components and genetic parameters of early growth traits in Luzhong mutton sheep estimated using different animal models, as well as the −2lnL and AIC values for each model and trait, are shown in Table 3. The model with the lowest AIC value was selected as the most appropriate model, so Model 4 (including direct additive and maternal genetic effects as random effects with covariance between them) was optimal for genetic parameter estimation of early growth traits in Luzhong mutton sheep. The estimated values of direct heritability for the BWT, WWT, 6WT, ADG1, and ADG2 were 0.156 ± 0.057, 0.547 ± 0.031, 0.653 ± 0.031, 0.531 ± 0.035, and 0.052 ± 0.046, respectively, while those of maternal heritability were 0.201 ± 0.100, 0.280 ± 0.047, 0.197 ± 0.053, 0.275 ± 0.052, and 0.081 ± 0.092, respectively. The results of the chi-square test analysis of different animal models are shown in Table 4. For the WWT, the chi-square test differences between Model 4 and Models 1, 2, and 3 and Model 6 and Models 1, 2, 3, and 5 were highly significant (*p* < 0.01). However, the differences between Model 5 and Models 2 and 3 and between Model 6 and Model 4 were not significant (*p* > 0.05) for all traits, whereas the differences between Model 4 and Model 3 reached the level of significance for all traits.

### 3.3. Correlation Analysis of Early Growth Traits

The genetic and phenotypic correlations between early growth traits estimated using Model 4 are shown in Figure 1. The genetic correlation between ADG2 and ADG1 was low (0.011), and the genetic correlation between ADG2 and WWT was negative (−0.180) compared to the other variables; the remaining traits showed moderate to strong positive genetic correlations (0.371–0.984). Highly significant (*p* < 0.01) positive correlations were found between all trait phenotypes. The highest genetic and phenotypic correlations were found between the WWT and ADG1, with coefficients of 0.984 and 0.916, respectively, and the lowest genetic and phenotypic correlations were found between the ADG1 and ADG2, with coefficients of 0.011 and 0.124, respectively.

### 3.4. Correlation Analysis of Early Growth Traits

The genetic trends and phenotypic trends of the early growth traits of Luzhong mutton sheep are shown in Figure 2. The average breeding value of the BWT showed an overall increasing trend. The genetic trends of the WWT, 6WT, and ADG1 were similar, with all of them decreasing and then increasing, and the average breeding value of the ADG2 showed an overall decreasing trend. The genetic trends were 0.0138 kg, 0.1076 kg, 0.1420 kg, 0.0012 kg/day, and 0.00002 kg/day, respectively; the phenotypic trends were 0.0132 kg, 1.7770 kg, 1.610 kg, 0.0006 kg/day, and −0.0011 kg/day, respectively.

## 4. Discussion

### 4.1. Comparison of Different Animal Models

In the present study, the highest direct heritability estimates were obtained for the BWT estimated using Model 1, suggesting that considering only direct additive genetic effects can bias the genetic parameter estimates. In contrast, the highest direct heritability estimates for the WWT, 6WT, ADG1, and ADG2 estimated using Model 4 may be due to the high negative covariance of direct additive genetic effects with maternal genetic effects. Among the genetic effects estimated using Model 4 and Model 6 for the BWT and ADG2, the maternal genetic effect was greater than the direct additive genetic effect, and the maternal genetic effect also had a greater proportion in other traits, which indicated that the maternal genetic effect had a greater influence on the accuracy of the estimation of the genetic parameters for the early growth traits in Luzhong mutton sheep. Chauhan et al. [13] reported that maternal genetic effects have a greater impact on the growth rate of Manali sheep. In this study, the maternal permanent environmental effect on the BWT was greater than that on the WWT, 6WT, and ADG1, and the effect decreased gradually over time. This suggests that the nutritional status and body condition of ewes have significant impacts on newborn lambs. These findings are similar to those found for Alpine Merino [14] and Mecheri [2] lambs. The results of this study suggest that maternal permanent environmental effects are related to the uterine condition, nutritional status in late gestation, and maternal behavior in ewes [15].

This finding was consistent with the results of the optimal model studies of Arabi [16] and Mecheri sheep [2]. In conclusion, Model 4 is the optimal model for estimating the genetic parameters of early growth traits in Luzhong mutton sheep, and maternal genetic effects have an important influence on early growth traits in Luzhong mutton sheep. The animal model used to estimate genetic parameters and breeding values in this study contained fixed, random, and residual effects. The male model is a simplified form of the animal model, which is superior to the male model both statistically and from the genetic breeding point of view, and can make fuller use of individual phenotypic and genealogical information. As far as maternal effects are concerned, they have been taken into account within the model. The sheep farms are not well equipped to classify the birth rank and rearing rank of lambs for data analysis, so this item was not included in the data collection process. The objective of the present study was to select lambs for early growth traits and to classify them for later rearing to improve breeding efficiency and save production costs.

### 4.2. Estimation of Heritability

Newborn weight is often used as an indicator of growth potential [2]. This study presents moderate heritability estimates for the BWT in Luzhong mutton sheep, with a direct heritability estimate of 0.156 ± 0.057 and a maternal heritability estimate of 0.201 ± 0.100. These estimates are similar to those reported by Fadili et al. [17] for Moroccan Timahdit sheep, Aguirre et al. [18] for Santa Ines sheep, and Mandal et al. [19] for Muzaffarnagari sheep.

The estimated heritability of body weight in lambs is relatively higher than those reported by Haile et al. [20] for Awassi sheep and Tesema et al. [21] for Dorper × indigenous crossbred sheep. However, this value is lower than those reported by Habtegiorgis et al. [22] for Doyogena sheep and Li et al. [14] for Alpine Merino sheep. These differences may be attributed to variations in sample size, species, and rearing environment. Lambs with a higher BWT are generally considered to have greater potential economic value due to their later growth and development. Additionally, improving the BWT of lambs can help reduce lamb mortality rates. Based on the estimated direct heritability of the BWT in medium Luzhong mutton sheep, this study suggests that there is room for improvement in this breed.

The direct heritability estimate of the WWT was 0.547 ± 0.031, and the maternal heritability estimate was 0.280 ± 0.047, indicating high and medium heritability, respectively. Similar to the present study, the WWT heritability estimates in Central Anatolian Merino sheep [23] in Turkey and Lohi sheep [24] in Pakistan were nearly the same. In contrast, some studies reported lower weaning heritability estimates in sheep, such as Jordanian Awassi sheep [25], Indian Mecheri sheep [26], and Moroccan Sardi sheep [27]. The results of the present study showed that the selection of lambs with higher body weights at weaning had a greater impact on improving growth performance in Luzhong mutton sheep. In addition, the medium maternal heritability of this trait indicated that the WWT was related not only to the genetic potential of the lambs themselves but also to the feeding ability and milk production of the mothers.

The direct heritability estimate of the 6WT was 0.653 ± 0.031, and the maternal heritability estimate was 0.197 ± 0.053, indicating that the 6WT was a high heritability trait, similar to the findings of Lalit et al. [28] in Harnali sheep. The direct heritability estimated by Solomon et al. [29] in Menz sheep was reported to be higher than that in Luzhong mutton sheep. However, the estimated heritability values in sheep breeds such as Nellore [30] and Doyogena [22] were lower than this result, which might be related to the differences between breeds. The higher estimated direct heritability of the 6WT indicated that the selection of Luzhong mutton sheep at 6 months of age could improve their growth performance. In addition, the direct heritability effect was much greater than the maternal heritability effect for the 6WT, showing that the influence of dams on lambs gradually decreased with increasing age.

The direct heritability and maternal heritability of the ADG1 in the present study were 0.531 ± 0.035 and 0.275 ± 0.052, respectively. The direct heritability estimates of the ADG1 were similar to those of the Madras Red [31], Munjal [32], and Harnali [28] sheep breeds. However, other studies have also reported relatively low estimates of direct heritability in the range of 0.06–0.27 [13,33,34]. The direct and maternal heritability values of ADG2 were 0.052 ± 0.046 and 0.081 ± 0.092, respectively, indicating low heritability. Similarly, lower direct heritability estimates in the range of 0.02–0.17 were observed in Munjal [32], Harnali [13,35], and Garole x Malpura (GM) [33] sheep breeds. The maternal heritability was lower for the WT than for the BWT and WWT, and maternal heritability for the ADG2 was lower than that for the ADG1, which suggests that as the age of the lambs increased, the maternal effect decreased.

In addition, the direct heritability of all traits estimated by the best model in this study was higher than the overall heritability, indicating that the early growth traits in Luzhong mutton sheep cannot be selected on the basis of phenotypic values alone; rather, the genetic performance of the traits should be selected to achieve the effect of selection for improvement. The differences between the results of the present study and those of previous studies may be related to differences in the characteristics of the studied breeds, climatic and geographical conditions, the level of feeding management, and the structure of the data used. Maniatis et al. [36] demonstrated that the structure of the data and the number of records of the ewes were important determinants in estimating the direct and maternal effects of early growth traits. While six different animal models were used in this study, each model contained different random effects, which made the data structure of each model different, thus affecting the estimation of the genetic parameters of early growth traits in Luzhong mutton sheep.

### 4.3. Genetic and Phenotypic Correlations

The estimated genetic correlations between different growth traits ranged from −0.180 to 0.984. Estimates using the best animal Model 4 showed highly significant positive genetic correlations (0.371–0.699) for the BWT, WWT, 6WT, ADG1, and ADG2. In the present study, the genetic correlation between the BWT and WWT was lower than those reported by Sharif et al. [24] for Lohi sheep (0.66), Oyieng [37] for Red Maasai sheep (0.54), and Hanford [38] for Columbia sheep (0.56) but was greater than that reported by Ahmad et al. [39] for Corriedale sheep (0.43). The results of this study suggest that selection for the BWT improves the growth of lambs in later stages of life, but selection may increase the chances of difficult births. A negative genetic correlation was found between the WWT and ADG2, while the genetic correlation among all other traits was positive, which suggests that selection for the WWT negatively affects the ADG2. This result may be because the reliance on and demand for breast milk by the lambs decreases in the later stages of growth, and the influence of maternal effects is significantly reduced; this may also be because the additive genetic variance increases as lambs grow older [28]. The genetic correlation between the ADG1 and ADG2 was extremely low, indicating that selecting for the daily weight gain at the birth to weaning stage does not have a significant effect on optimizing the next stage of growth and fails to meet the selection and breeding objectives. The results also showed that selection of Luzhong mutton sheep for these two stages had little effect on the other stages.

The correlations between all the phenotypes of the Luzhong mutton sheep were positive. Genetic correlations between traits are influenced by a large number of factors, while correlations between phenotypes depend on common environmental factors [3]. In this study, the genetic correlation values between traits were greater than the phenotypic correlation values, except for the correlations between the ADG1 and ADG2, 6WT and WWT, and ADG2 and WWT. In addition, BWT and 6WT were positively correlated with other traits in Luzhong mutton sheep; thus, selection based on these two traits would result in positive selection for other traits.

### 4.4. Genetic and Phenotypic Trends

In the present study, both genetic and phenotypic trends of early growth traits showed fluctuating trends and were generally consistent among the traits. The mean breeding value of the BWT was lowest in 2015, reached a maximum in 2017, then started to decline and increase again after 2018, with an overall increasing trend (0.0138 kg). This is greater than the genetic variation in the BWT estimated by Habtegiorgis et al. [22] for Doyogena sheep (0.00085 kg), Areb et al. [40] for Bonga sheep (0.002 kg) and Haile et al. [20] for Awassi sheep (0.0001 kg). Direct genetic trends of negative BWT in some sheep breeds, such as Baluchi (−0.13 kg) [41] and Doyogena (−0.0026 kg) [22], have also been reported in the literature. The genetic trends of the WWT, 6WT, and ADG1 were similar, all showing a decrease followed by an increase, with the average breeding values reaching a minimum in 2016 and then increasing sharply, with an overall increasing trend. The genetic trend coefficients were 0.1076 kg/day, 0.1420 kg/day, and 0.0012 kg/day, respectively, which were greater than the results of Areb et al. [40] for Bonga sheep. Compared with those in other studies, the genetic trend coefficients for the WWT and 6WT were higher in Luzhong mutton sheep than in Makooei, Doyogena, and Dorper sheep breeds [6,22,42]. Direct ADG1 genetic trend coefficients were reported to be lower in Awassi (0.000103 kg) [20] and Bonga (0.000317 kg) [40] sheep than in the present study, whereas Tesema et al. [21] reported similar but negative ADG1 genetic trend coefficients in Dorper × indigenous crossbred sheep populations (−0.001103 kg/day), which may be related to the breed of sheep. The mean ADG2 breeding values, on the other hand, showed an overall decreasing trend (−0.00002 kg/day), with Areb et al. [40] reporting a greater and positive ADG2 genetic trend coefficient (0.000485 kg/day). The mean phenotypic values showed an overall increasing trend after decreasing to the lowest value in 2016, except for the ADG2, which decreased to its lowest value in 2017 and then showed an overall increasing trend. In contrast to the findings of Besufkad et al. [42] and Kariuki et al. [43], who reported a decreasing phenotypic trend in Dorper sheep over the years, the above results suggest that Luzhong mutton sheep can be selected at the early growth stage of 6 months of age, which in turn will shorten the fattening cycle and improve breeding efficiency.

In this study, we compared the estimation effects of different animal models for the early growth traits Luzhong mutton sheep, and Model 4, which considered the direct additive genetic effect, maternal genetic effect, and its covariance, was the optimal model for estimating the genetic parameters of the early growth traits of Luzhong mutton sheep. The fixed effects included in Model 4 were direct additive genetic effects, maternal genetic effects, and genetic covariance between the two. The model also has some limitations. First, it contains fewer fixed effects and does not take into account the interactions between the fixed effects, and the residuals are larger. Second, the data collected in this study were mainly focused on the breeding base of Luzhongton mutton sheep. At present, the breed has been widely promoted in China, so the scope of data collection should be increased in the later stage, and the model should continue to undergo optimization for the early growth traits of this breed.

## 5. Conclusions

To improve the accuracy of genetic parameter estimation and to develop a more accurate and perfect breeding program, the influence of maternal effects should not be neglected. The high direct heritability of Luzhong mutton sheep, a new meat breed specifically bred for housed feeding, suggests that selection efforts have been successful in recent years. In addition, the high heritability estimate of the 6WT in Luzhong mutton sheep suggested that selection based on this trait could lead to improved genetic gains.

## Figures and Tables

**Figure 1 animals-14-01754-f001:**
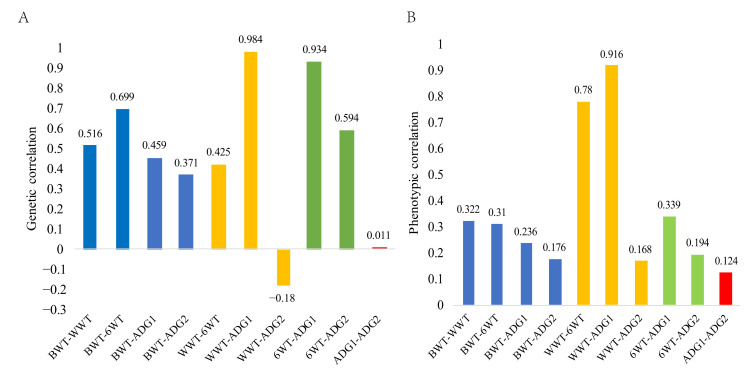
Genetic and phenotypic correlation of early growth traits in Luzhong mutton sheep. (**A**) Genetic correlation estimates between growth traits of Luzhong Mutton Sheep. (**B**) Phenotypic correlations between growth traits of Luzhong Mutton Sheep. Abbreviations: BWT, birth weight; WWT, weaning weight; 6WT, six-month weight; ADG1, average daily gain from birth to weaning; ADG2, average daily gain from weaning to six months. A moderate to strong genetic association exists between the weaning weight and other body weight traits, and a strong genetic relationship exists between subsequent traits.

**Figure 2 animals-14-01754-f002:**
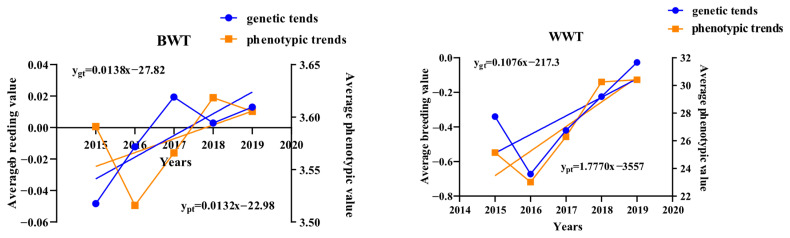
Genetic and phenotypic trends of early growth traits in Luzhong mutton sheep.

**Table 1 animals-14-01754-t001:** Descriptive statistics of growth trait data for Luzhong mutton sheep.

Traits	BWT (kg)	WWT (kg)	6WT (kg)	ADG1 (kg)	ADG2 (kg)
N	2464	2923	2428	2424	1836
Mean ± Standard error	3.589 ± 0.009	27.395 ± 0.101	41.231 ± 0.114	0.276 ± 0.001	0.152 ± 0.001
Standard Deviation	0.435	5.472	5.593	0.049	0.027
Min	2.500	11.200	28.500	0.120	0.050
Max	4.400	37.500	54.000	0.372	0.259

Abbreviations: BWT, birth weight; WWT, weaning weight; 6WT, six-month weight; ADG1, average daily gain from birth to weaning; ADG2, average daily gain from weaning to six months.

**Table 2 animals-14-01754-t002:** Effects of non-genetic factors on growth traits of Luzhong mutton sheep.

Traits	Birth Year			Birth Season			Sex		
	Mean Square	*F*	*p*	Mean Square	*F*	*p*	Mean Square	*F*	*p*
BWT	0.934 **	5.37	0.0003	0.707 **	407.00	0.0068	33.107 **	190.44	<0.0001
WWT	5746.681 **	292.03	<0.0001	649.839 **	33.02	<0.0001	1933.506 **	98.25	<0.0001
6WT	3770.473 **	181.93	<0.0001	1193.097 **	57.57	<0.0001	461.788 **	22.28	<0.0001
ADG1	0.282 **	165.42	<0.0001	0.040 **	23.49	<0.0001	0.122 **	71.44	<0.0001
ADG2	0.007 **	6.42	<0.0001	0.005 **	4.20	0.0057	0.008 **	7.07	0.0079

Abbreviations: BWT, birth weight; WWT, weaning weight; 6WT, six-month weight; ADG1, average daily gain from birth to weaning; ADG2, average daily gain from weaning to six months; *F*, F value. ** indicates an extremely significant difference (*p* < 0.01).

**Table 3 animals-14-01754-t003:** Estimation of variance components and genetic parameters for the early growth traits of Luzhong mutton sheep using different animal models.

Traits	Model	σ_a_^2^	σ_m_^2^	σ_a,m_	σ_c_^2^	σ_e_^2^	σ_p_^2^	σ_a_^2^/σ_p_^2^	σ_m_^2^/σ_p_^2^	σ_c_^2^/σ_p_^2^	σ_e_^2^/σ_p_^2^	h_T_^2^	−2lnL	AIC
BWT	1	0.036				0.138	0.174	0.209 ± 0.035			0.791	0.209	−1834.122	−1830.12
	2	0.034			0.003	0.134	0.171	0.200 ± 0.059		0.015 ± 0.042	0.785	0.200	−1813.992	−1807.99
	3	0.031	0.032			0.122	0.185	0.168 ± 0.062	0.170 ± 0.055		0.661	0.253	−1813.832	−1807.83
	**4**	**0.028**	**0.036**	**−0.014**		**0.128**	**0.177**	**0.156 ± 0.057**	**0.201 ± 0.100**		**0.723**	**0.136**	**−1839.796**	**−1831.80**
	5	0.027	0.017		0.005	0.128	0.176	0.152 ± 0.094	0.096 ± 0.047	0.026 ± 0.147	0.726	0.200	−1811.112	−1803.11
	6	0.025	0.031	−0.012	0.003	0.128	0.176	0.143 ± 0.078	0.177 ± 0.209	0.020 ± 0.160	0.727	0.132	−1830.646	−1820.65
WWT	1	7.599				8.309	15.907	0.478 ± 0.027			0.522	0.478	10,801.161	10,805.16
	2	7.551			0.063	8.295	15.909	0.475 ± 0.046		0.004 ± 0.030	0.521	0.475	10,801.139	10,807.14
	3	7.560	0.051			8.298	15.909	0.475 ± 0.045	0.003 ± 0.029		0.522	0.477	10,801.145	10,807.15
	**4**	**8.544**	**4.372**	**−5.310**		**8.026**	**15.633**	**0.547 ± 0.031**	**0.280 ± 0.047**		0.513	0.177	10,750.433	10,758.43
	5	7.373	0.267		0.045	8.248	15.932	0.463 ± 0.122	0.017 ± 0.081	0.003 ± 0.083	0.518	0.471	10,801.139	10,809.14
	6	8.538	4.023	−5.250	0.296	8.024	15.631	0.546 ± 0.127	0.257 ± 0.121	0.019 ± 0.113	0.513	0.171	10,750.416	10,760.42
6WT	1	8.944				8.861	17.805	0.502 ± 0.031			0.498	0.502	9257.474	9261.47
	2	8.977			0.003	8.833	17.813	0.504 ± 0.054		0.0002 ± 0.034	0.496	0.504	9257.733	9263.73
	3	8.969	0.009			8.834	17.812	0.504 ± 0.052	0.0005 ± 0.032		0.496	0.504	9257.622	9263.62
	**4**	**11.434**	**3.442**	**−5.857**		**8.483**	**17.502**	**0.653 ± 0.031**	**0.197 ± 0.053**		**0.485**	**0.250**	**9208.665**	**9216.66**
	5	8.926	0.032		0.002	8.852	17.811	0.501 ± 0.126	0.002 ± 0.078	0.0001 ± 0.081	0.497	0.502	9257.994	9265.99
	6	11.409	3.441	−5.817	0.034	8.458	17.524	0.651 ± 0.139	0.196 ± 0.114	0.002 ± 0.106	0.483	0.251	9208.734	9218.73
ADG1	1	6.13 × 10^−4^				7.89 × 10^−4^	1.40 × 10^−3^	0.437 ± 0.028			0.563	0.437	−13,536.32079	−13,532.32
	2	6.06 × 10^−4^			6.700 × 10^−6^	7.89 × 10^−4^	1.40 × 10^−3^	0.432 ± 0.051		0.005 ± 0.035	0.563	0.432	−13,534.7641	−13,528.76
	3	5.93 × 10^−4^	3.22 × 10^−5^			7.82 × 10^−4^	1.41 × 10^−3^	0.421 ± 0.052	0.023 ± 0.036		0.556	0.433	−13,532.18564	−13,526.19
	**4**	**7.28 × 10^−4^**	**3.77 × 10^−4^**	**−5.10 × 10^−4^**		**7.76 × 10^−4^**	**1.371 × 10^−3^**	**0.531 ± 0.035**	**0.275 ± 0.052**		**0.565**	**0.111**	**−13,586.49552**	**−13,578.50**
	5	6.04 × 10^−4^	1.47 × 10^−5^		8.000 × 10^−7^	7.85 × 10^−4^	1.405 × 10^−3^	0.430 ± 0.140	0.010 ± 0.100	0.001 ± 0.103	0.559	0.435	−13,534.12069	−13,526.12
	6	7.20 × 10^−4^	3.74 × 10^−4^	−5.02 × 10^−4^	1.300 × 10^−6^	7.78 × 10^−4^	1.371 × 10^−3^	0.525 ± 0.141	0.273 ± 0.136	0.001 ± 0.133	0.567	0.112	−13,586.32817	−13,576.33
ADG2	1	1.16 × 10^−5^				7.25 × 10^−4^	7.36 × 10^−4^	0.016 ± 0.039			0.984	0.016	−11,269.75683	−11,265.76
	2	2.19 × 10^−5^			4.92 × 10^−5^	6.75 × 10^−4^	7.46 × 10^−4^	0.029 ± 0.049		0.066 ± 0.041	0.905	0.029	−11,277.21275	−11,271.21
	3	1.20 × 10^−5^	3.76 × 10^−5^			6.93 × 10^−4^	7.43 × 10^−4^	0.016 ± 0.056	0.051 ± 0.049		0.933	0.041	−11,275.12621	−11,269.13
	**4**	**3.92 × 10^−5^**	**6.04 × 10^−5^**	**−2.51 × 10^−5^**		**6.74 × 10^−4^**	**7.48 × 10^−4^**	**0.052 ± 0.046**	**0.081 ± 0.092**		**0.900**	**0.042**	**−11,279.22475**	**−11,271.22**
	5	2.42 × 10^−5^	4.06 × 10^−5^		5.50 × 10^−6^	6.77 × 10^−4^	7.48 × 10^−4^	0.032 ± 0.041	0.054 ± 0.126	0.007 ± 0.132	0.906	0.060	−11,278.20752	−11,270.21
	6	3.61 × 10^−5^	6.66 × 10^−5^	−1.99 × 10^−5^	5.90 × 10^−6^	6.62 × 10^−4^	7.51 × 10^−4^	0.048 ± 0.050	0.089 ± 0.195	0.008 ± 0.157	0.882	0.053	−11,275.76864	−11,265.77

Abbreviations: σ_a_^2^, direct additive genetic variance; σ_m_^2^, maternal additive genetic variance; σ_a,m_, individual and maternal genetic covariance; σ_c_^2^, maternal permanent environmental variance; σ_e_^2^, residual variance; σ_p_^2^, phenotypic variance; σ_a_^2^/σ_p_^2^, direct additive genetic effects; σ_m_^2^/σ_p_^2^, maternal genetic effect; σ_c_^2^/σ_p_^2^, maternal permanent environmental effect; σ_e_^2^/σ_p_^2^, residual effect; h_T_^2^, total heritability; lnL, log-likelihood function; AIC, Akaike information criterion, the best fitted model according to AIC is shown in bold type.

**Table 4 animals-14-01754-t004:** Chi-square test analysis of different animal models.

Model	BWT	WWT	6WT	ADG1	ADG2
2:1	−20.130 ^ns^	0.022 ^ns^	−0.259 ^ns^	−1.557 ^ns^	7.456 **
3:1	−20.290 ^ns^	0.016 ^ns^	−0.148 ^ns^	−4.135 ^ns^	5.369 *
4:1	5.674 ^ns^	50.728 **	48.809 **	50.175 **	9.468 **
5:1	−23.010 ^ns^	0.022 ^ns^	−0.519 ^ns^	−2.200 ^ns^	8.451 *
6:1	−3.4756 ^ns^	50.745 **	48.740 **	50.007 **	6.012 ^ns^
4:2	25.804 **	50.706 **	49.068 **	51.731 **	2.012 ^ns^
5:2	−2.880 ^ns^	0.000 ^ns^	−0.261 ^ns^	−0.643 ^ns^	0.995 ^ns^
6:2	16.654 **	50.723 **	48.999 **	51.564 **	−1.444 ^ns^
4:3	25.964 **	50.712 **	48.957 **	54.310 **	4.099 *
5:3	−2.720 ^ns^	0.006 ^ns^	−0.372 ^ns^	1.935 ^ns^	3.081 ^ns^
6:3	16.815 **	50.729 **	48.888 **	54.143 **	0.642 ^ns^
6:4	−9.150 ^ns^	0.017 ^ns^	−0.069 ^ns^	−0.167 ^ns^	−3.456 ^ns^
6:5	19.535 **	50.723 **	49.260 **	52.207 **	−2.439 ^ns^

Abbreviations: BWT, birth weight; WWT, weaning weight; 6WT, six-month weight; ADG1, average daily gain from birth to weaning; ADG2, average daily gain from weaning to six months. ** indicates an extremely significant difference (*p* < 0.01); * indicates a significant difference (*p* < 0.05); ^ns^ indicates no difference (*p* > 0.05).

## Data Availability

The data sets used in this study belong to the Institute of Animal Science and Veterinary Medicine, Shandong Academy of Agricultural Sciences. The data sets of the current study will be made available on reasonable request with permission from the related government agency.

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
