# Peer review of "Estimation of Genetic Parameters for Early Growth Traits in Luzhong Mutton Sheep"

_animals, 2024, doi:10.3390/ani14121754_

Round 1
Reviewer 1 Report
Comments and Suggestions for Authors
Dear authors, first of all I want to emphasize that the topic of your article is particularly relevant and corresponds to the stage at which the breeding program for the Luzhong breed is being implemented. However, I would like to ask you some questions and make some recommendations that could improve the quality of your article and thus also serve for further refinements of the selected linear models. First, you should insert more information about the Luzhong breed into the text. As you write, it is widespread in Shandong province, but you need to state what the population size and trend is in the last 5 years. An important issue essential to the application of a linear model for the estimation of genetic parameters is to describe the production systems in which the Luzhong breed is grown. What is the number of flocks in which the breeding program will be applied? In material and method, you must describe the method of insemination of the ewes: natural insemination or artificial insemination? In the model you described, there is no flock effect included!? On the other hand you describe on line 90 that herds are grouped by gender and age? This should be described in detail to get an idea of the structure of the environmental effects! If the data you are processing comes from only one flock, then your conclusions are not valid for the whole breed but only for that flock! Another important question that arises is how the lambs are weaned, how do you weigh the lambs and do you adjust the weaning weight depending on the age of weaning? You must describe and cite the software used to estimate the genetic parameters - the heritability coefficient and the repeatability coefficient!? In addition to the Log-likelihood function and the Akaike information criterion, the Bayesian information criterion (BIC)is usually calculated? Could you add BIC as well? In the Material and Method section, you should describe how do you calculate the genetic trend! The notes made do not detract from your scientific results, but are intended to improve the article!
Author Response
Please refer to the annex

Reviewer 2 Report
Comments and Suggestions for Authors
This was an interesting an well-written manuscript. The use of extensive lamb growth data, multiple models, and a well defined model selection criteria provided a systematic approach to determining the model of best fit for estimating the genetic parameters for early growth traits in Luzhong mutton sheep.
More information about the management of the flocks would be useful to the reader (perhaps provided as supplementary information). For example were the animals housed or on pasture or a mixture of both? What is the dietary formula? At what stages - during pregnancy, during winter? How is weigh of lambs before their first feed ensured? Are the animals lambed inside? .
In regards to the models, it would be interesting to know why paternal (sire) effects were not considered and why birth and/or rearing rank were not included in any of the models. An explanation as to why these were not included would be appropriate in the methods section. Also in the methods section (or as a supplementary table) it would also be useful for the assumption for each model and its implications to be acknowledged. For example y=Xb+Z1a+Z2m+e Cov(a,m)≠0; the assumption is the possibility of correlation or covariance between the genetic effects (direct additive and maternal) on the phenotype of the offspring; The model implies that the direct additive and maternal genetic effects may not act independently on the trait being studied. Explicitly detailing this information will make the article more accessible to those who are not so statistically inclined and who are not familiar with what the equations mean.
In the discussion it would be useful to acknowledge the limitation of the optimal model (model 4) and discuss other parameters that could have been included eg sire effects, birth rank and rearing rank effects but that the modeling process aims to achieve a balance between model complexity and goodness of fit. In fact, it would be appropriate to have a limitations section in the discussion that acknowledges all the limitations of the study.
Please see the attached file for minor edit suggestions.

Minor editing of English language required as indicated in the annotated PDF
Author Response
Please refer to the annex
